# Palladium Nanoparticles Incorporated Fumed Silica as an Efficient Catalyst for Nitroarenes Reduction via Thermal and Microwave Heating

Afaf Y. Khormi [1], Badria M. Al-Shehri [1,2,3], Fatimah A. M. Al-Zahrani [1,*], Mohamed S. Hamdy [1], Amr Fouda [4,*] and Mohamed R. Shaaban [5,6]

[1] Chemistry Department, Faculty of Science, King Khalid University, P.O. Box 9004, Abha 61413, Saudi Arabia
[2] Research Center for Advanced Materials Science (RCAMS), King Khalid University, P.O. Box 9004, Abha 61413, Saudi Arabia
[3] Unit of Bee Research and Honey Production, Faculty of Science, King Khalid University, P.O. Box 9004, Abha 61413, Saudi Arabia
[4] Department of Botany and Microbiology, Faculty of Science, Al-Azhar University, Nasr City, Cairo 11884, Egypt
[5] Department of Chemistry, Faculty of Science, Cairo University, Giza 12613, Egypt
[6] Department of Chemistry, Faculty of Applied Science, Umm Al-Qura University, Makkah Almukaramah 21514, Saudi Arabia
* Correspondence: falzhrani@kku.edu.sa (F.A.M.A.-Z.); amr_fh83@azhar.edu.eg (A.F.)

**Abstract:** The reduction of nitroarenes to arylamines is a synthetically important transformation both in the laboratory and in industry. Herein, Palladium (Pd) nanoparticles were synthesized via incorporation with mesoporous fumed silica material by doping technique. Water was used as a solvent and the as-synthetized material was reduced by using $NaBH_4$ to ensure the total transformation of PdO into Pd nanoparticles. The synthesized sample was characterized by using inductively coupled plasma (ICP) elemental analysis, X-ray powder diffraction (XRD), $N_2$ sorption measurement, scanning electron microscope (SEM), energy-dispersive spectroscopy (EDX), and transmission electron microscopy (TEM). Data showed that the Pd nanoparticles were successfully synthesized and supported on the mesoporous silica with an average size in the ranges of 10–20 nm, with an irregular shape. The purity of the synthesized sample was confirmed by EDX analysis which exhibits the presence of Si, O, and Pd. The catalytic activity of the prepared sample was evaluated in the heterogeneous reduction of nitroarenes to aromatic amines. Reduction reaction was monitored by Shimadzu GC-17A gas chromatography (GC, Japan) equipped with flam ionization detector and RTX-5 column, 30 m × 0.25 mm, 1-μm film thickness. Helium was used as carrier gas at flow rate 0.6 mL/min. Interestingly, the green hydrogenation of nitroarenes to primary amine compounds was achieved in an aqueous solution with high efficiency and in a short time; moreover, the reusability of heterogeneous $Pd-SiO_2$ was performed for four repeated cycles with more than 88% of efficiency at the fourth run. Finally, the heterogeneity of catalysis with high reliability and eco-friendly processes is a super new trend of nitroarenes reduction in the industry and economic scales.

**Keywords:** palladium; silica; heterogeneous; catalysis; microwave

## 1. Introduction

Azoxy aromatic compounds, in addition to the azo organic compounds, are important in the industrial scale and in pharmaceutical applications which take place in the preparation of a wide range of different organic dyes, colored food additives, optical storage media, medical therapy reagents, and several types of therapy derivatives [1]. The direct coupling to producing amino organic compounds from nitro aromatics is considered challenging when setting up azo aromatic compounds and azoxy aromatic compounds with controlled selectivity (without forming amines) at ambient temperature [2,3]. Functionalized aromatic

amines are important chemicals and intermediates for the synthesis of agrochemicals, herbicides, pharmaceuticals, agricultural chemicals, surfactants, polymers, and dyes. The growing demand for functionalized amines has fueled the development of environmentally friendly and cost-effective synthetic methods for their scalable production. One of the most straightforward and industrially applicable methods for synthesizing anilines is the sustainable reduction of aromatic nitro compounds [4], despite significant progress in the direct hydrogenation of nitroarenes using pressurized $H_2$ as a hydrogen source with non-noble metal catalysts [5–7]. Transfer hydrogenation could achieve safe reductions under mild conditions without the need for specialized reaction setups; as a result, $NaBH_4$, $HCOONH_4$, HCOOH, hydrazine hydrate, silane, and $H_3PO_3$ have emerged as potential hydrogen storage materials for nitro compound reduction [8,9].

Over the past few decades, the nanoparticles (NPs) of noble metals have been noted to be heavily involved in the field of organic transformation, owing to their unique structures and high activity, and the development of Pd NPs is of great advantage [10,11]. Numerous methods have been developed to prepare these catalysts, which have been widely used in hydrogenation [12–15], hydrogenolysis [16,17], decarbonylation [18,19], and coupling reactions [20–23]. Recyclable catalysts are very favorable for the reduction of nitroaromatic compounds and several transition metals (e.g., Pd, Pt, Rh, Ru, etc.) are reported to be efficient for this process [24,25]. Heterogeneous catalysts play a critical role in the global production of fuels.

Heterogeneous catalysts are critical for the global production of both bulk fuels and chemicals as well as fine chemicals. Because of the world's growing population, rising global demand for energy and feedstock, a push for sustainability, and limited availability of rare and noble metals, the development of new and better catalysts is critical. New catalysts with high selectivity and selectivity must be developed [26]. Ideally, such advancement is accomplished through the creation of custom-made systems for specific applications. Today's heterogeneous catalysts are frequently supported by noble or transition metal systems and solid acids such as zeolites. These catalysts are used in a variety of reactions, including cracking, hydrogenation, and oxidation, particularly when bulk chemicals are produced on a large scale. In comparison to homogeneous catalysts, heterogeneous catalysts have the advantages of easy separation from the product, high stability, and good recyclability [27,28].

Heterogeneous nanomaterial-based catalysts (nanocatalysts) have recently attracted much attention in catalytic reactions because of their high activities and the benefit of accessibility to specific sites, where carbon–carbon coupling, oxidation, or reduction reaction as organic reactions are beginning to produce implausible results by using heterogenous catalysis [29,30]. The zeolite type in addition to doped noble metals are considered the primary types of heterogenous catalysts used from decades ago as heterogenous catalysts in the reduction of nitro organic compounds in catalyzed reactions [31].

Recent research has revealed that supported metal nanoparticles can be used to generate amino organic compounds through heterogenous catalysis. The heterogeneous catalysts, including palladium (Pd), palladium–platinum, and Rhodium nanoparticles on a different solid matrix of supports, including different metal oxides, have proven their capacity to generate the target product of amino organic arenes free of side-products; hence, the reductive product will be the main product produced by these solid nanoparticles that are imbedded into nanostructures of metal oxides support such as $Al_2O_3$ and $SiO_2$ [22].

Mesoporous silica (e.g., MCM-41, MCM-48, SBA-15, and TUD-1) has attracted increasing attention to be considered an important class of nanostructured support materials [32,33]. The large surface area, well-defined porous architecture, and their ability to incorporate metal atoms within the mesopores serve as a promising support material for designing a variety of different catalysts [32,34].

Recently, several studies have been reported for the formation of aniline compounds via heterogeneous catalysis using supported metal nanoparticles, as shown in Table 1. The heterogeneous catalysts, such as Pd nanoparticles on various supporters, such as $TiO_2$, $SiO_2$,

PVP–iron powder, g-$C_3N_4$, and $Fe_3O_4$, proved to be effective in producing quantitative yields of anilines and without the formation of self-coupled products [31,32].

**Table 1.** Comparison of catalytic performance for the reduction with Pd and other catalysts.

| Entry | Catalyst | Reductant | Solvent | Yield (%) | Reference |
|-------|----------|-----------|---------|-----------|-----------|
| 1 | Pd-Au/$TiO_2$ | $NaBH_4$ | $H_2O$ | 76 | [35] |
| 2 | Pd/$SiO_2$ nanospheres | $NaBH_4$ | $H_2O$ | 15 | [35] |
| 3 | Pd/PVP–iron powder | $NaBH_4$ | $H_2O$ | 99 | [36] |
| 4 | Pd/g-$C_3N_4$ | HCOOH | $H_2O$ | 99 | [37] |
| 5 | Pd–Pt/$Fe_3O_4$ (1 mol%) | $NH_3BH_3$ | MeOH | 96 | [38] |
| 6 | Pd-$SiO_2$ | $NaBH_4$ | $H_2O$ | 100 | Current study |

Therefore, in the current research, an attempt was performed to replace conventional homogeneous Pd complexes with a heterogeneous catalyst. To reach this goal, Pd-NPs were incorporated into fumed mesoporous silica by using water as the only solvent. The properties of Pd-$SiO_2$ were characterized by using several physical and chemical techniques; moreover, the prepared catalyst was investigated in the hydrogenation of nitroarenes to primary amine compounds by using two different heating methods.

## 2. Results and Discussion

### 2.1. Synthesis of the Catalyst

Noble NPs are highly appealing candidates due to their distinct combination of physical, chemical, mechanical, and structural properties [39,40]. Many developments in this area continue to fascinate the materials research community, and can be broadly classified as chemical sensors, biosensors, Förster resonance energy transfer (FRET), and microelectronic applications. Herein, Pd-NPs were synthesized through incorporation with fumed mesoporous silica by using water as the only solvent. Several physical and chemical characterization techniques were applied to investigate the chemical and morphological structure of the prepared catalyst.

2.1.1. Inductively Coupled Plasma Spectroscopy (ICP)

First, inductively coupled plasma (ICP) elemental analysis showed that the Pd content in the final solid product is 0.415%, while the intended content was 0.5%. This result showed that the loading efficiency is almost 83% by using only water as a solvent; moreover, the textural properties of the prepared Pd-$SiO_2$ are compared with the bare silica sample in Table 2.

**Table 2.** The textural properties of bare $SiO_2$ and Pd-$SiO_2$.

| Sample | Surface Area ($m^2$/g) | Pore Volume ($cm^3$/g) | Pore Diameter (nm) |
|--------|------------------------|------------------------|--------------------|
| $SiO_2$ | 224.5 | 0.148 | 2.431 |
| Pd-$SiO_2$ | 268.4 | 0.121 | 2.187 |

2.1.2. $N_2$ Adsorption/Desorption Isotherms

The surface area of the sample was increased from 224 to 268 $m^2$/g as a result of increasing the surface area of the Pd nanoparticles, while the pore volume and the pore diameter did not change significantly.

2.1.3. X-ray Diffraction (XRD) Technique

The crystallinity of the prepared and as-synthetized samples was compared with that of the bare $SiO_2$ by using the XRD technique. Figure 1 presents the XRD patterns of the three samples, the bare $SiO_2$, the as-synthetized Pd-$SiO_2$ (before reduction), and the final solid Pd-$SiO_2$ samples. The bare $SiO_2$ sample exhibited one broad peak centered at 2θ of 22.5° as an indication of the amorphous nature of the commercial $SiO_2$ sample. After incorporating Pd and before reducing the sample, the peaks of PdO at 2θ values of

33.7° and 42.8°. These peaks are attributed to the planes of (101) and (110) of palladium oxide species according to JCPDS card no. 43-1024 [41]. Pd-SiO$_2$ sample after reduction showed different peaks at 2θ values of 40.0°, 46.5°, and 67.9°, which can be referred to the planes of (111), (200), and (220) of Pd nanoparticles according to the JCPDS card no. 89-4897 [42]. The XRD data demonstrate that the reduction process completely reduced the PdO nanoparticles in the as-synthetized samples to Pd$^0$ nanoparticles in the final solid product.

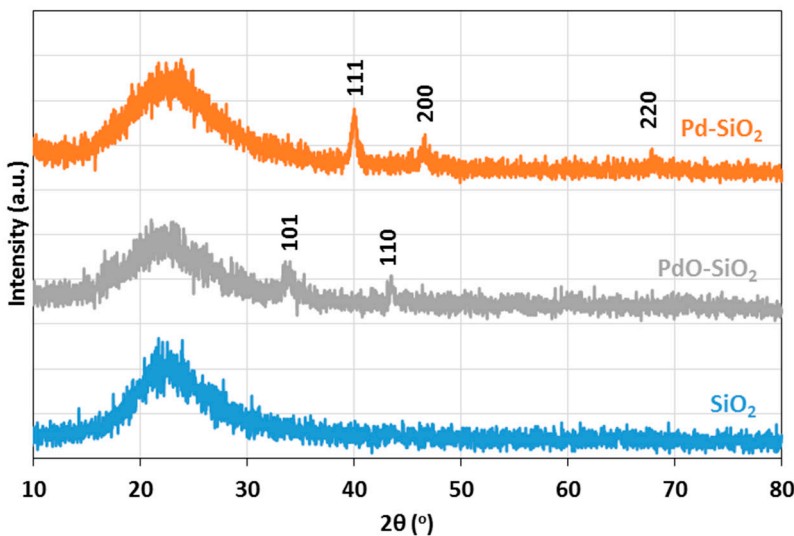

**Figure 1.** The XRD pattern of the bare SiO$_2$, the as-synthesized PdO-SiO$_2$, and the final solid Pd-SiO$_2$ samples.

### 2.1.4. Scanning Electron Microscopy (SEM)

The morphological structure of the Pd-SiO$_2$ sample was investigated by using scanning electron microscopy equipped with EDX analysis. The micrographs of the sample are presented in Figure 2A,B. The particles of Pd-SiO$_2$ exhibited the irregular shape of bulky amorphous silica; moreover, the surface of the sample was clean, and no crystalline particles of Pd were observed as an indication of the total incorporation of Pd nanoparticles in the silica framework. The corresponding EDX analysis showed only the elements of Si, O, and Pd, which are present in the sample without any other elements as an indication of the purity of the sample (Figure 2C). Finally, the elemental mapping (Figure 2D) showed a well distribution for the Pd nanoparticles throughout the entire sample.

The high area of solid supports of the surface, such as the mesoporous silica, is considered one of the specific control factors to establish the status of catalytic performance of the hydrogenation of the aromatic nitro group and is classified as a promising support of catalytic reduction [43]. From other side, the porosity degree of the matrix assists significantly in the binding of doped noble metals' nanoparticles in the matrix structure; in addition, it assists the passing of organic molecules through to the intra channels of the meso-type pore structure; nevertheless, the mesoporous silica are reported as highly stable materials thermally and hydrothermally in oxygen and moist air conditions; furthermore, these properties might be shifted by controlling different parameters such as the power of hydrogen (pH), silica precursor, type of template, calcination, and thermal temperature; therefore, the developing of metals' nanoparticles as heterogenous catalysts which are eco-friendly, cost-effective, and with high performance and stability is considered critical in organic reactions.

Moreover, the distribution of the Pd nanoparticles in the silica matrix was examined by TEM analysis. The obtained micrograph is presented in Figure 3. The micrograph clearly shows well distribution of Pd nanoparticles with a size of 10–20 nm throughout the silica framework.

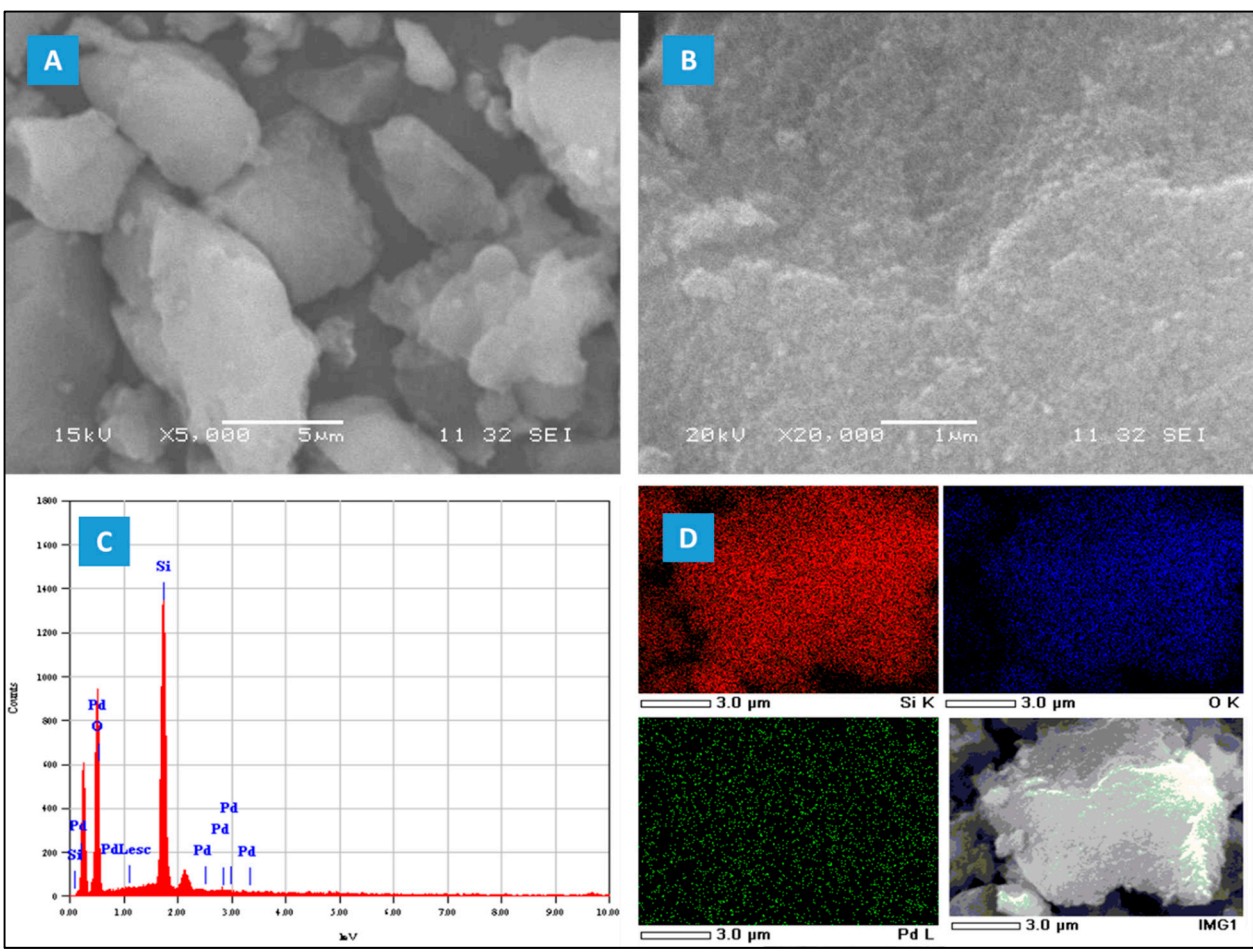

**Figure 2.** SEM analysis of the prepared Pd-SiO$_2$ sample. (**A,B**) are the micrographs of the sample with different magnifications, (**C**) EDX analysis, and (**D**) the elemental mapping of the sample.

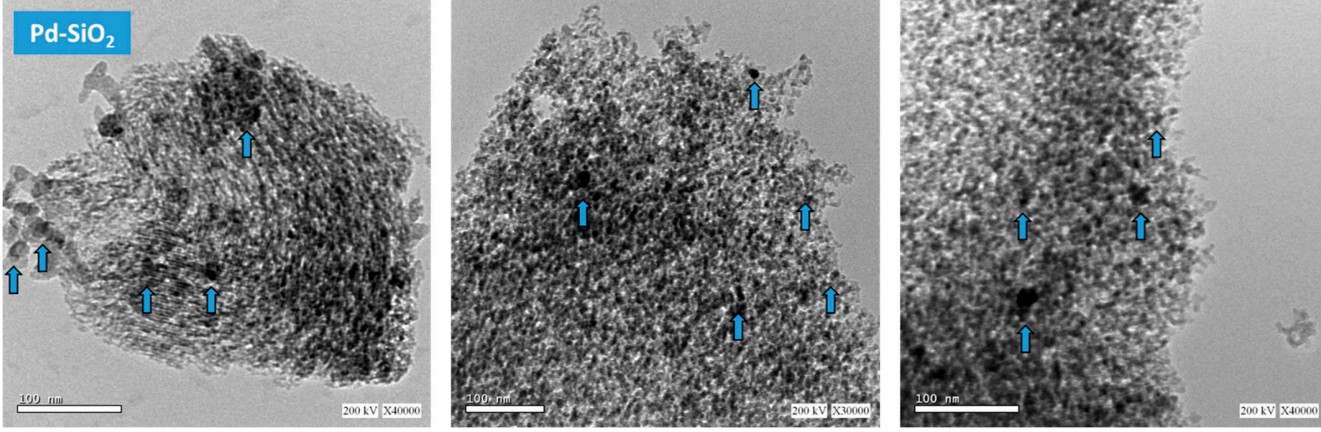

**Figure 3.** Different TEM images for the Pd-SiO$_2$ sample; arrows are pointing to the Pd nanoparticles.

### 2.2. Catalytic Activity

Nitroaromatic compounds are few among the most common organic pollutants and explosives, and, hence, their conversion to some value-added products is of great relevance and importance [44,45]. The catalytic efficiency of the Pd-SiO$_2$ has been examined in order to get full conversions of nitroarenes to aromatic amines with high yields in aqueous media; also, the heating mode for the reaction is one of the most important parameters in this work. Pd-SiO$_2$ was found to be an efficient catalyst for the reduction of nitroarenes with

excellent yield using water as a solvent regardless of the nature of the substituent pendant on the aromatic ring.

The reusability and durability of the catalysts are very important criteria for practical applications [46]. We examined the recycling and reusability of the Pd nanoparticles confined in mesoporous silica for the reduction of nitrobenzene. The catalyst was successfully separated by centrifugation and reused for four consecutive cycles of the reduction of nitrobenzene with more than 88% of efficiency at the fourth cycle (see Figure 4).

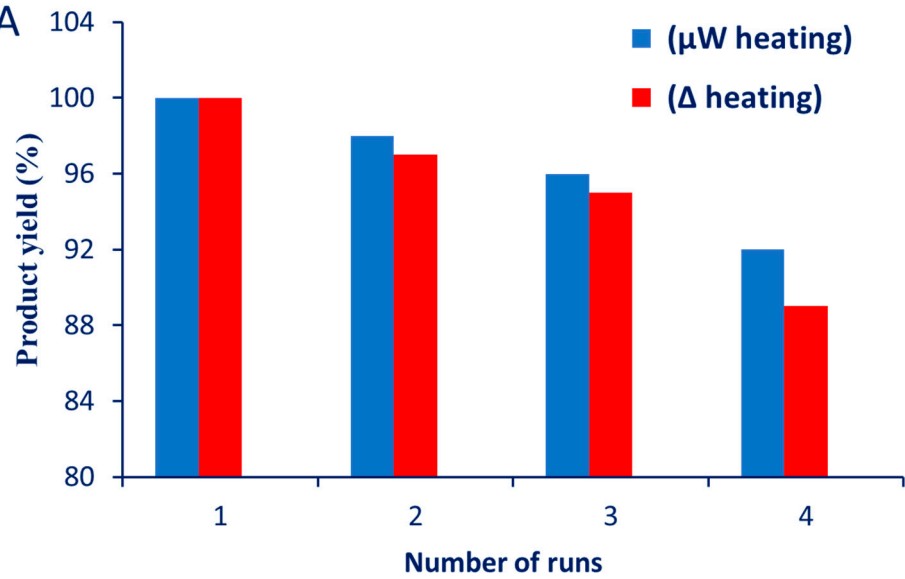

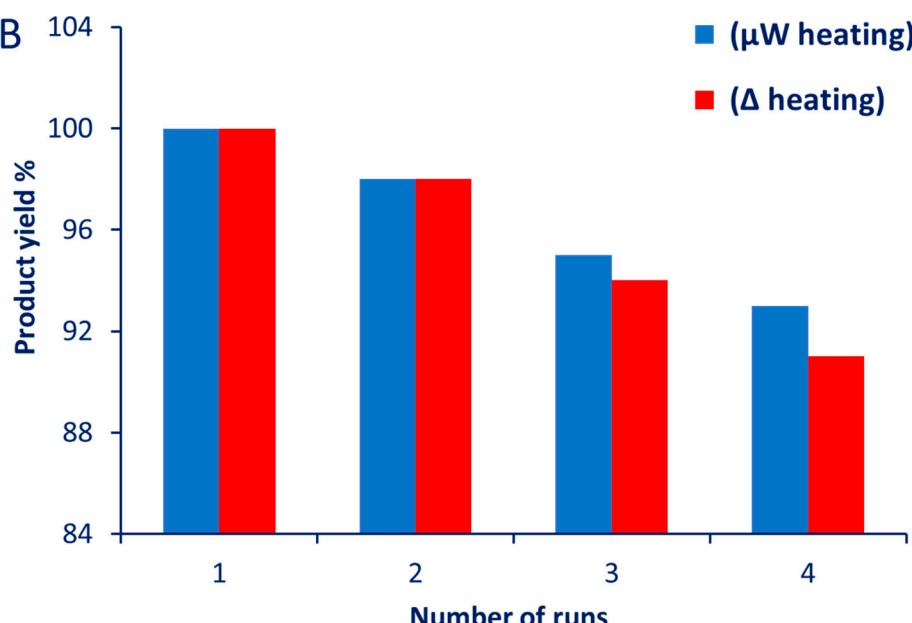

**Figure 4.** Reusability of Pd-SiO$_2$ in the reduction reaction of (**A**) 4-nitrobenzylalcohol and (**B**) 4-nitrophenylhydrazine.

Although, of these results, the structural nature of the catalysts played a critical role in the catalytic performance of the reactions. The porosity of the catalyst markedly changes the course of the organic reaction—specially, stability, activity, and selectivity are inevitably affected by the heterogenous catalyst as a solid support in the reduction of nitroarenes; therefore, the research hot points focused on the framework in order to disclose the ability

of support applied to physiochemical stability and activity, in addition to studying the role of surface area value in the provision of the high numbers of active sites in the organic reaction; consequently, the nanoparticles of silica-loaded noble metals have been attractive as eco-friendly nano-solid catalysts whether on an industrial or research scale due to the solvent-free protocol of amino arenes reduction.

### 2.3. Catalytic Performance

The reduction of the nitroaromatic compounds to aniline derivate compounds with Pd–SiO$_2$–NaBH$_4$ was fast and the reaction was completed in 1–3 min in a microwave unit. It gave 100% conversion with a single product (see Scheme 1, Table 3). The complete conversion (100%) was confirmed by GC analysis.

**1a  R: CH2OH**

**1b  R: NHNH2**

**Scheme 1.** Reaction scheme for the reduction of nitroaromatic compounds in water using Pd–SiO$_2$.

**Table 3.** Conventional and microwave-assisted reduction reaction of nitroaromatic compound.

| Compd. no. | Δ Heating | | μW Heating | |
|:---:|:---:|:---:|:---:|:---:|
| | Time (h) | Yield % | Time (min) | Yield % |
| **1a** | 2 | 100 | 1 | 100 |
| **1b** | 3 | 100 | 3 | 100 |

When Pd-SiO$_2$/NaBH$_4$ reactions were carried out under microwave influence, a 100% conversion of nitrobenzylalcohol and nitrophenylhydrazine was observed. This demonstrates that palladium boride on silica is an efficient system for selectively reducing nitroaromatics to aromatic amines. The yield of amine compounds was detected using GC analysis. During our research on in situ generated palladium boride on silica, we discovered that the aromatic ring of nitrobenzene remained intact under reaction conditions; unlike when Raney Ni was used as a catalyst, the reaction could proceed to aromatic ring hydrogenation [47].

When sodium borohydride was added, palladium particles adsorbed on the surface of the silica support, facilitating the formation of Pd boride. The rate of hydrogenation appears to be dependent on the rate of NaBH$_4$ decomposition to form Pd boride.

Electrons can be transferred from boron to palladium, resulting in metallic palladium that is electron-rich and boron that is electron-deficient [48,49]; as a result, the catalyst activity and selectivity of the product are greatly enhanced. The electronic interaction between the N=O groups in nitro aromatics and the metallic active sites could be one of the plausible adsorption mechanisms—the forward donation of electrons from the N=O bonding's highest occupied molecular orbital (HOMO), i.e., from the N=O to the dz2 and s orbits of the metallic Pd atom, and the back donation of electrons from the metallic Pd atom's dx$^2$-y$^2$ orbit to the lowest unoccupied molecular orbital (LUMO), i.e., *N=O. Because *N=O is an anti-bonding orbit, increased back electron donation to the *N=O caused by the high electron density on the Pd active sites can also activate the N=O bond and promote hydrogenation [48].

All the results and analytical data identify that the Pd, in association with silica, is the active species and major contributor to maximizing the activation of sodium borohydride to form palladium boride on silica support to transfer hydrogen.

## 3. Materials and Methods

### 3.1. Materials and Chemicals

All chemicals, including solvents, are spectroscopic grade and purchased from Sigma-Aldrich, Makkah Almukaramah, Saudi Arabia, and used without further purification. 4-nitrobenzyl alcohol and 4-nitro phenyl hydrazine were purchased from Sigma-Aldrich, Saudi Arabia.

### 3.2. Equipments and Tools

Melting points were measured on a Gallenkamp melting point apparatus (current is 0.3 A and voltage is 220–240 V, 50 Hz). Microwave irradiation was conducted on a CEM mars machine (CEM Microwave Technology Ltd., Buckinghamshire County, United Kingdom). CEM has several vessel types that are designed for their ovens: closed-system vessels including the HP-500 (500 psig material design pressure and 260 °C), with liners composed of PFA, and these are ideal for many types of samples. HP-500 Plus vessels are ideal for routine digestion applications. We processed up to 14 high-pressure vessels per run with temperatures up to 260 °C or pressures up to 500 psi. Reduction reaction was monitored by Shimadzu GC-17A gas chromatography (GC, Tokyo, Japan) equipped with flame ionization detector and RTX-5 column, 30 m $\times$ 0.25 mm, 1-$\mu$m film thickness. Helium was used as carrier gas at a flow rate 0.6 mL/min. Samples were withdrawn from the reaction mixture periodically. Injection volume was 1 $\mu$L, and total flow was 100 mL/min. $^1$H NMR spectra were recorded using a Bruker Avance 850 instrument (850 MHz for $^1$H, 125 MHz for $^{13}$C, Taipei City, Taiwan), and a Varian Mercury VXR-300 NMR spectrometer (International Equipment Trading Ltd., Mundelein, IL, USA) was used for products in DMSO-$d_6$.

### 3.3. Synthesis

Commercial fumed silica (OS 995, 99.9% Ottokemi, Mumbai, India) was used as a support. In a typical synthesis procedure, 1 g of silica was suspended in water and then a stoichiometric amount of palladium (II) nitrate dihydrate ($Pd(NO_3)_2 \cdot 2H_2O$, 98% Sigma-Aldrich) was added to the suspension under vigorous stirring and was kept overnight. The suspension was filtrated and the doped silica material was heated in the furnace at 400 °C for 4 h to obtain the as-synthetized $PdO-SiO_2$; finally, the as-synthetized sample was reduced by using $NaBH_4$ to form the Pd nanoparticles. The final product was kept in a sealed clean glass tube and labeled as $Pd-SiO_2$.

### 3.4. Catalyst Characterization

#### 3.4.1. Inductively Coupled Plasma Spectroscopy (ICP)

The amount of Pd found in the produced material was measured using the ICP technique utilizing a Thermo Scientific ICAP 7000 series unit, model number 1,340,910 (Thermo Fisher Scientific Inc., Waltham, MA, USA), equipped with Qtegra software, on a QuantaChrome NOVA 2000e device (Boynton Beach, FL, USA) [50].

#### 3.4.2. N$_2$ Adsorption/Desorption Isotherms

Nitrogen sorption isotherms were measured at liquid nitrogen temperature using a QuantaChrome Autosorb-6B instrument (Boynton Beach, FL, USA). The Brunauer, Emmett and Teller (BET) technique was employed to determine specific surface areas and the pore size distributions and resolved from adsorption branches of the N2-sorption curves using the Barrett–Joynor–Halenda (BJH) model [51].

### 3.4.3. X-ray Diffraction (XRD) Technique

The produced samples' crystallinity was examined using the X-ray diffraction (XRD) technique. XRD was performed using a Shimadzu 6000 DX diffractometer (Austin, TX, USA) that has a graphite monochromator and a CuK (=0.1541 nm) radiation source [52].

### 3.4.4. Scanning Electron Microscopy (SEM)

The morphological characteristics of the amino-functionalized mesoporous silica and catalyst were examined by scanning electron microscopy (SEM), Jeol Model 6360 LVSEM, USA, on a JEOL JEM-2100 electron microscope (Tokyo, Japan) using a field emission gun as the electron source [53]. Transmission electron microscopy (TEM, a Philips CM30UT) was performed. Energy-dispersive X-ray (EDX, Jeol Model 6360 LVSEM, Portland, OR, USA), was used to produce qualitative and semi-quantitative elemental analysis.

### 3.4.5. High-Resolution Transmission Electron Microscopy HR-TEM

The structural properties of the samples were examined using an electron microscope (JEM-2200FS, accelerating voltage 200 kV) in the transmission of high-resolution electron microscopy mode (PFEM, high-resolution transmission electron microscopy—HRTEM). A model of the microscope JEOL JEM-2200FS showed an accelerating voltage of 200 kV. Resolution: by points—0.19 nm; on the lattice—0.1 nm; in the mapping mode—0.2 nm; and in the HAADF mode—0.14 nm. For analysis of the elemental composition, electron microscopy was used in energy-dispersive X-ray spectroscopy mode (EMF, energy-dispersive X-ray spectroscopy (EDS)) [54].

### *3.5. Catalytic Activity*

3.5.1. General Procedure for Reduction of Nitroarenes Catalyzed by $Pd-SiO_2$ in Water under Conventional Heating

A mixture of nitroaromatic compound (1 mmol), $NaBH_4$ (5 mmol), $Pd-SiO_2$ (0.05 g), and 10 mL distilled water was refluxed for the appropriate time. The progress of the reaction was monitored by GC (Tokyo, Japan). The reduction of nitroarene was achieved in the general atmosphere. After the reduction reaction was completed, the catalyst was separated from the solution by centrifugation and washed with water and ethyl acetate. The product was extracted from the reaction solution with ethyl acetate, then the extracts were dried using anhydrous sodium sulfate. The yield of the isolated product was obtained after evaporating ethyl acetate.

3.5.2. General Procedure for Reduction of Nitroarenes Catalyzed by $Pd-SiO_2$ in Water under Microwave Irradiation

A mixture of nitroaromatic compound (1 mmol), $NaBH_4$ (5 mmol), $Pd-SiO_2$ (0.05 g), and 10 mL distilled water was mixed in the specified CEM reaction vessel HP-500. The mixture was heated with microwave irradiation at 90 °C and 300 watts for the appropriate time. After the reduction reaction was complete (monitored by GC), the catalyst was separated from the solution by centrifugation and washed with water and ethyl acetate. Extraction of the products was accomplished by ethyl acetate, then the extracts were dried using anhydrous sodium sulfate. The yield of the isolated product was obtained after evaporating ethyl acetate.

*4-Aminobenzyl alcohol*: Compound **2a** (Scheme 2) is prepared according to the general procedure; **[1]H NMR** (850 MHz, DMSO-d_6, TMS): 8.17 (dd, 2H, Ar-H), 7.53 (dd, 2H, Ar-H), 5.36 (t, 1H, OH), 4.58 (d, 2H, $NH_2$), and 4.62 (d, 2H, $-CH_2$). **[13]C NMR** 62.59, 62.99, 20.27, 122.27, 122.84, 125.48, 127.08, 127.23, 127.60, 142.65, 145.14, 146.76, 147.53, and 151.33. **IR** OH-Stretch 3518.58, N-H stretch primary amine at 3111.62, 3080.55, C=C-H Stretch 2923.98, C-H Stretch aliphatic 2869.32, N-H bending primary amine 1602.11, and C-N Stretch at 1247.

**Scheme 2.** Structure of 4-Aminobenzyl alcohol.

*4-Hydrazinyl aniline*: Compound **2b** (Scheme 3) is prepared according to the general procedure; **¹H NMR** (850 MHz, DMSO-d$_6$, TMS): 7.51 (d, 2H, Ar-H), 7.00 (t, 1H, NH), 6.60 (d, 2H, Ar-H), 6.54 (d, 2H, NH$_2$), and 5.75 (S, 2H, NH$_2$). **¹³C NMR** 22.49, 113.92, 114.07, 114.32, 116.08, 124.24, 126.08, 129.15, 129.26, 143.60, and 151.44. **IR** N-H stretch primary amine at 3353.89, 3214.81, C=C-H Stretch 3034.30, N-H bending primary amine 1621.03, and C-N Stretch at 1240.

**Scheme 3.** Structure of 4-Hydrazinyl aniline.

## 4. Conclusions

The obtained results of the current research showed that Pd-SiO$_2$ can replace the conventional homogeneous Pd complexes in the hydrogenation of nitroarenes to primary amine compounds. Pd nanoparticles (with an average size of 10–20 nm) were incorporated in a commercial SiO$_2$ matrix by using water as a solvent. The morphology and chemical content of repaired heterogenous catalyst was confirmed by several characteristic techniques, whereas the XRD data demonstrate that the reduction process completely reduced the PdO nanoparticles in the as-synthetized samples to Pd$^0$ nanoparticles in the final solid product; therefore, the results exhibited uniform dispersion of palladium oxide within silica matrix. The perfect size of embedded nanoparticles was one of the factors that led to the high performance of the solid catalyst; furthermore, the high specific area (almost to 270 cm$^2$/g) assists in the steady reduction of nitroarenes; so, the prepared heterogeneous catalyst of palladium nanoparticles imbedded into commercial silica matrix was found to be an efficient and highly active catalyst for the heterogeneous reduction reaction of nitroarenes to aromatic amines using aqueous solution as a medium under conventional heating conditions of heating under 90 °C for the appropriate time and microwave irradiations under 90 °C and 300 watts for the appropriate time. The reusability of the palladium into SiO$_2$ heterogeneous catalyst was performed four times with nitrobenzene with no significant loss of activity and high stability. Palladium boride generated in situ by the decomposition of sodium borohydride on palladium silica reduced various nitro aromatics to aromatic amines with good conversions and selectivity in a short duration in the range of one to three minutes in microwave conditions. In most cases, 100% conversions were achieved; whereas, the reduction reaction gas chromatography (GC-MS with the using of Helium gas as carrier gas at flow rate 0.6 mL/min. Samples were withdrawn from the reaction mixture periodically; therefore, cost-effective, eco-friendly, reusable, and stable Pd-SiO$_2$ can be considered a promising catalyst for organic transformations in the industrial and environmental applications, such as the reduction of nitro aromatic compounds to amino aromatic products.

**Author Contributions:** Conceptualization, A.Y.K., B.M.A.-S., F.A.M.A.-Z., M.S.H. and M.R.S.; methodology, A.Y.K., B.M.A.-S., F.A.M.A.-Z., M.S.H. and M.R.S.; software, A.Y.K., B.M.A.-S., F.A.M.A.-Z., M.S.H., A.F. and M.R.S.; validation, A.Y.K., B.M.A.-S., F.A.M.A.-Z., M.S.H., A.F. and M.R.S.; formal analysis, A.Y.K., B.M.A.-S., F.A.M.A.-Z., M.S.H. and M.R.S.; investigation, A.Y.K., B.M.A.-S., F.A.M.A.-Z., M.S.H. and M.R.S.; resources, A.Y.K., B.M.A.-S., F.A.M.A.-Z., M.S.H., A.F. and M.R.S.; data curation, A.Y.K., B.M.A.-S., F.A.M.A.-Z., M.S.H. and M.R.S.; writing—original draft preparation, A.Y.K., B.M.A.-S., F.A.M.A.-Z., M.S.H. and M.R.S.; writing—review and editing, A.Y.K., B.M.A.-S., F.A.M.A.-Z., M.S.H., A.F. and M.R.S.; visualization, A.Y.K., B.M.A.-S., F.A.M.A.-Z., M.S.H. and M.R.S.; supervision, A.Y.K., B.M.A.-S., F.A.M.A.-Z., M.S.H. and M.R.S.; project administration, A.Y.K., B.M.A.-S., F.A.M.A.-Z., M.S.H., A.F. and M.R.S.; funding acquisition, A.Y.K., B.M.A.-S., F.A.M.A.-Z., M.S.H., A.F. and M.R.S. All authors have read and agreed to the published version of the manuscript.

**Funding:** This research was funding by the Scientific Research at the King Khalid University through the Small Groups Project (under grant number GRP/21/43).

**Data Availability Statement:** The data presented in this study are available on request from the corresponding authors.

**Acknowledgments:** The author extends their appreciation to the Deanship of Scientific Research at the King Khalid University for funding this work through the Small Groups Project (under grant number GRP/21/43).

**Conflicts of Interest:** The authors declare no conflict of interest.

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
