# Peer review of "Palladium Nanoparticles Incorporated Fumed Silica as an Efficient Catalyst for Nitroarenes Reduction via Thermal and Microwave Heating"

_catalysts, doi:10.3390/catal13020445_

Round 1

Reviewer 1 Report

First of all, I would to congratulate the authors for submitting their manuscripts to this journal.

Khormi et al., in their research article “Palladium Nanoparticles Incorporated Fumed Silica as an Efficient Catalyst for Nitroarenes Reduction via Thermal and Microwave Heating” have shown the importance of incorporating Pd NPs in fumed silica for remediation for Nitroarenes. The manuscript is updated, and scientifically sound but I have found several issues in the manuscript after going through it. My recommendations are

1.     In the abstract section authors have directly used abbreviations like SEM, EDS etc. without introducing them for the very first time.

2.     Objectives are not clear. It must be clearly mentioned at the end of the introduction section.

3.     In line no 96, ICP is used without abbreviation. Please introduce it first, so that readers could understand it easily.

4.     Fig.3 TEM image is very fuzzy and ugly. Provide a better image.

5.     Line 174, in section 3.1, “furtherpurification” needs space.

6.     Try to prove the chemical grade, company, city and country of origin for all the chemicals.

7.     In section 3.3.Characterization, line 196  “Qtegra Soft wear”. I guess it should be software.

8.     City and country must be proved for all the instruments.

9.     Rewrite the conclusion. The first sentence must be rephrased.

Conclusion: Major revision

Author Response

Dear reviewer, thank you very much for your valuable comments. We revised the manuscript as the reviewers suggested and we answered all reviewer comments point by point as shown in author's response.

Reviewer 2 Report

The manuscript entitled “Palladium Nanoparticles Incorporated Fumed Silica as an Efficient Catalyst for Nitroarenes Reduction via Thermal and Microwave Heating” is quite interesting, well framed, but the manuscript still needs some Minor corrections before publishing in this journal.

I advise the authors to consider the following points when revising their manuscript.

1.     In line 54 and 55, check correct the typo errors.

2.     Line 58-59- “The properties of Pd-SiO2 were characterized several physical and chemical techniques” changed to “The properties of Pd-SiO2 were characterized by several physical and chemical techniques.”

3.     The caption (2.1 the prepared catalyst) to appropriate caption.

4.     Line 74, correct the word “surface are” to “surface area”.

5.     In Figure 1, SiO2 & PdO-SiO2 to be written as “SiO2” and PdO-SiO2.

6.     Line 79 change “tree” to three. Also line 223- “unhydrous” changed to anhydrous

7.     Also check the entire manuscript for typo and scientific errors. (such as temperature degree notations, 2 theta notations etc.)

8.     In the reduction of nitroarene experiment, does the reactions are performed in general atmosphere or N2 atmosphere?

9.     The author mentioned that the nitro-aromatic compound was 100 % converted. How the author confirmed the conversion?

10.  How the amine compounds yield was calculated? Did column chromatography performed or Crude NMR yield or some other technique? The procedure for obtained yield should be included in the manuscript.

11.  In the synthesis of catalyst, the author mentioned that “Finally, the as-synthetized sample was reduced by using NaBH4 to form the Pd nanoparticles. The final product was kept in a sealed clean glass tube and labeled as Pd-SiO2.”

But in the General procedure for reduction of nitroarenes catalyzed by Pd-SiO2 in water section, for synthesis “A mixture of nitro-aromatic compound (1 mmol), NaBH4 (5 mmol), Pd-SiO2 (0.05 g) and 10 ml distilled water was refluxed for appropriate time.”

Does the author used NaBH4 two times? One time in the conversion of as-synthetized  PdO-SiO2 to Pd-SiO2 and in the second time in the reduction of nitroarenes ?.

Author Response

(The authors gave the same response as above.)

Reviewer 3 Report

The author has reported manuscrooi[t with Palladium Nanoparticles Incorporated Fumed Silica as an Effi- 2 cient Catalyst for Nitroarenes Reduction via Thermal and Mi- 3 crowave Heating.  The article is well written and rcommended after major changes, 

1. abstract revision and write in numerical form

2. Why TEM has only 1?

3. The results are not very impressive, there is need to modified and need to optimized fyther facot. 

Author Response

(The authors gave the same response as above.)

Round 2

Reviewer 1 Report

The quality of the manuscript improved significantly.

It should be accepted in its current form.